# Roles of the Unsaturated Fatty Acid Docosahexaenoic Acid in the Central Nervous System: Molecular and Cellular Insights

**DOI:** 10.3390/ijms23105390

**Published:** 2022-05-12

**Authors:** Ana B. Petermann, Mauricio Reyna-Jeldes, Lorena Ortega, Claudio Coddou, Gonzalo E. Yévenes

**Affiliations:** 1Departamento de Fisiología, Facultad de Ciencias Biológicas, Universidad de Concepción, Concepción 4070386, Chile; anapetermann@udec.cl; 2Millennium Nucleus for the Study of Pain (MiNuSPain), Santiago 8330025, Chile; mauricio.reyna@ucn.cl (M.R.-J.); lorena.ortega@ucn.cl (L.O.); 3Departamento de Ciencias Biomédicas, Facultad de Medicina, Universidad Católica Del Norte, Coquimbo 1781421, Chile; 4Núcleo para el Estudio del Cáncer a Nivel Básico, Aplicado y Clínico, Universidad Católica del Norte, Antofagasta 1270709, Chile

**Keywords:** docosahexaenoic acid, DHA, GPR40, cell survival, neural morphology, synaptic function, CNS tumors

## Abstract

Fatty acids (FAs) are essential components of the central nervous system (CNS), where they exert multiple roles in health and disease. Among the FAs, docosahexaenoic acid (DHA) has been widely recognized as a key molecule for neuronal function and cell signaling. Despite its relevance, the molecular pathways underlying the beneficial effects of DHA on the cells of the CNS are still unclear. Here, we summarize and discuss the molecular mechanisms underlying the actions of DHA in neural cells with a special focus on processes of survival, morphological development, and synaptic maturation. In addition, we examine the evidence supporting a potential therapeutic role of DHA against CNS tumor diseases and tumorigenesis. The current results suggest that DHA exerts its actions on neural cells mainly through the modulation of signaling cascades involving the activation of diverse types of receptors. In addition, we found evidence connecting brain DHA and ω-3 PUFA levels with CNS diseases, such as depression, autism spectrum disorders, obesity, and neurodegenerative diseases. In the context of cancer, the existing data have shown that DHA exerts positive actions as a coadjuvant in antitumoral therapy. Although many questions in the field remain only partially resolved, we hope that future research may soon define specific pathways and receptor systems involved in the beneficial effects of DHA in cells of the CNS, opening new avenues for innovative therapeutic strategies for CNS diseases.

## 1. General Aspects of Fatty Acids

In the last decades, attention has been driven towards different types of fatty acids (FAs) due to their multiple functions on health and disease. From a dietary perspective, FA are mainly incorporated in our diet as lipids, which are esterified forms between different FA and organic alcohols, such as cholesterol, sphingosine, or glycerol [1,2]. FAs contribute to plasma membrane fluidity and are critical molecules regulating cellular communication and intracellular signaling [1,3,4]. At the physiological level, FAs control a large variety of processes, including blood pressure, inflammation, central nervous system (CNS) function regulation, among others [2,3,4,5,6,7].

Free fatty acids (FFAs), as the non-esterified form of FA obtained after lipid hydrolysis by lipases [7], are saturated or unsaturated aliphatic chains with an even number of carbon atoms. FFAs possesses a carboxyl group at one end, and a free methyl group at the opposite side [3,4,7,8,9]. Saturated FFAs are composed only of single bonds, while unsaturated FFAs contain at least one double bond in their structure. Unsaturated FFAs can be subclassified as either monounsaturated (MUFA, one carbon double bond) or polyunsaturated (PUFA, more than one carbon double bond) according to their structures. With regard to the latter, PUFA are subclassified as ω-3 and ω-6 if their first double bond is on the third and sixth carbon counting from the ω-carbon, respectively [3,4]. Examples of ω-3 PUFA are alpha-linolenic acid (ALA), eicosapentaenoic acid (EPA), and docosahexaenoic acid (DHA) [4,10]. Other prominent PUFAs are docosapentaenoic acid-omega 6 (DPA-ω-6) and arachidonic acid (ARA), the latter being one of the major PUFAs in the mammal brain that seems to have a prominent role in brain development [2,11]. Interestingly, alterations in cellular processes involving FAs have been linked to many CNS disorders, including depression, schizophrenia, Alzheimer and Parkinson diseases, and cancer [6,12,13,14]. Among the FAs, DHA has been widely recognized as a key molecule for neuronal function and signaling, as well as a major PUFA in the mammalian brain [2,6,12,13,15,16].

## 2. An Overview on DHA Effects in the CNS

DHA (C_22_H_32_O_2_) is a long-chain PUFA formed by 22 carbon atoms, six cis-double bonds and a terminal carboxyl moiety [4]. Mammals obtain a major proportion of their required DHA from food. High quantities of DHA are found in seafood, especially on krill oil and fatty fish, among other varieties [17]. Breast milk is another, but rather limited, source of dietary DHA. Although DHA can be endogenously synthesized from ALA, this conversion rate represents around 0.5% of total DHA amounts in humans [4,7,15,18], having negligible effects on plasma DHA in adults [19]. Hepatic DHA synthesis in rats fed with diets containing 4.6% ALA as the only ω-6 PUFA source are capable of maintaining DHA levels in the brain [20]. This ALA to DHA conversion is also enough to meet brain DHA requirements in human adults, which are estimated to be around 2.4–3.8 mg/day [21] However, this amount could be insufficient for fetal and infant neurodevelopment in humans; therefore, a dietary supplementation of ≈200 mg DHA/day or ≈250–375 mg EPA + DHA/day has been recommended for lactating and pregnant women. This dosage is also recommended for cardiovascular disease prevention in adults [21,22]. DHA has multiple roles on the CNS, both structural and functional under physiological conditions, but also as a promising agent against CNS diseases. Some of these effects of DHA on the CNS are summarized on Figure 1.

DHA, as a PUFA, is particularly enriched in the CNS and the peripheral nervous system (PNS), being part of glycerophospholipids of these systems, normally esterified to the position 2 of the glycerol backbone [2]. DHA is the main PUFA within the brain [4,23,24]. In nerve cells, one of the central roles of DHA is to maintain plasma membrane integrity [15,25,26]. Neuronal membranes are primarily formed by oleic acid (OLA), arachidonic acid (ARA), several esters of phosphatidylinositol, and DHA, which is the main esterified FA contained in phosphatidylethanolamine (PE) and phosphatidylserine (PS) [27,28,29,30]. Phospholipid analysis of cortical neuron membranes incubated for 24–48 h with 25 µM DHA showed an increase in DHA content in both PE and PS, and in the intracellular concentration of these phospholipids [31]. Other experiments have shown that esterified DHA is concentrated at high amounts on neuronal membranes and synaptic vesicles [29]. In line with their abundance, other lines of research have shown that reduction in DHA levels impairs neurogenesis, neurite growth, and neurotransmission [30,32,33]. Interestingly, studies using dietary DHA supplementation have shown benefits in many pathological conditions, such as depression, Alzheimer’s disease, tumor growth, and pain [12,13,34,35,36]. Regarding the latter condition, linoleic acid derivatives have been described in rodent and human models as mediators with prospective functions promoting pain transmission, nociceptive sensitization, and scratching behavior in mice [37]. DHA supplementation has been associated with pain reduction in rat models treated with sodium valproate [38], and in patients affected with rheumatoid arthritis treated concomitantly with Janus kinase inhibitors, such as tofacitinib and baricitinib [39]. In clinical setups for the evaluation of migraine, increased EPA + DHA and reduced ω-6 FA diets were associated with a reduction in the frequency and severity of headaches [40].

In relation to the structural effects on membranes, DHA has a crucial role in visual function, where rod outer segment membranes showed high DHA contents, which have the role of maintaining disc shape in photoreceptor cells in the retina. The ω-3 FA deficiency in animal models correlated with impaired retinal responses to light and reduced a-wave amplitude on electroretinograms, these effects being able to be restored with ALA-enriched fat diets [2,41,42]. Additionally, supplementation with a nutraceutical formulation of DHA triglyceride, xanthophylls, vitamins, and minerals in patients with no proliferative diabetic retinopathy for 90 days was able to improve macular sensitivity and integrity in the early stages of this pathology [43]. It has been reported that ω-3 deficiency results in an altered response to light, as was observed in rats fed with fat-free diets [44]. Other studies have found that rats with diets deficient in alpha linoleic acid show reduced retinal responses to light [45]. When rhodopsin is extracted from normal or alpha linoleic acid-deficient rat retinas, a reduced capacity for photon absorption was found in the latter, suggesting an explanation for the diminished retinal sensitivity in ω-3 deficient rats [46]. Deficiencies in alpha linoleic acid resulted in reduced concentrations of DHA in the retina and the visual processing center of the occipital cortex [46]. Studies in vesicles containing rhodopsin, transducin and phosphodiesterase revealed that ω-3 restriction not only decreased rhodopsin activation, but also impaired the formation of rhodopsin-transducin complexes and phosphodiesterase activity. In addition to its role in visual function, it has been demonstrated that ω-3 deficiency has many consequences, such as macular degeneration [47], photoreceptor degeneration [48], or generation of a pro-inflammatory phenotype in retina [49]. Conversely, DHA improves neuroprotection, decreasing apoptosis in mice retina [50] and protecting against visible light-induced retinal damage [51], Therefore, DHA could have a potential therapeutic application against retinopathies [52]. Interestingly, other critical mammalian sensory systems have been found to be dependent, at least in part, on dietary ω-3 FA intake. For example, ω-3-FA-deficient rats showed alterations in auditory function and olfactory performance [53,54,55].

An additional finding related to ω-3 FA effects in cognition was that learning defects in mice reared for three generations on ω-3 deficient diets could be reversed not only by the addition of ω-3 fatty acids (ALA) to the diet, but also by a COX-2 inhibitor [56]. These data might be explained by other data, which reported that omega-3 deficiency significantly increased the activity, protein concentration, and mRNA expression of arachidonic acid regulatory phospholipase A2 (PLA2) (calcium-dependent cPLA2 and secretory sPLA2), and COX-2 in rat frontal cortex [57], and the expression of cPLA2, COX-2, and PGE2 in rat hypothalamus [58], thus presumably increasing the availability of free arachidonic acid for metabolism to PGE2.

Aside from its function in cell membranes, DHA is also considered a key molecule in cell signaling. Several studies have reported that dietary DHA intake influences brain glucose uptake, likely through the modulation of the glucose transporter GLUT-1 and Na^+^/K^+^ ATPase [2,59,60,61,62]. DHA is an essential precursor of many pro-resolving mediators, such as protectins and resolvins, which are signaling molecules involved in inflammation resolution [63]. On the other hand, DHA can be metabolized to N-docosahexaenoylethanolamide (DEA), also called synaptomide, which is the main endogenous ligand of the GPR110 receptor in the brain [64,65]. In addition, DHA may contribute to cell signaling through direct actions on receptors and ion channels [6,66]. In this context, the GPR40/FFAR1 receptor has emerged as a potential key mediator for the direct actions of DHA [6,30,67]. GPR40/FFAR1 is an FFA receptor expressed in different CNS regions, such as the brain cortex, hypothalamus, and spinal cord, among others [6,30,36]. Saturated and unsaturated FA have been identified as potential ligands of GPR40/FFAR1. DHA has been characterized as a direct agonist for GPR40/FFAR1, displaying apparent affinities lower than 10 µM in diverse cell types [6,68,69,70]. Evidence suggests that GPR40/FFAR1 is a mediator of DHA effects in different physiological processes, including insulin secretion, control of hormonal secretion on the gut, taste processing and bone remodeling [6,30]. In the CNS, GPR40/FFAR1 activation has been linked to pain control as well as to regulation of cognition and emotional behavior. For example, experiments with mice demonstrated that GPR40/FFAR1 activation or inhibition modulated pain behaviors [36,71,72].

Despite growing knowledge concerning the physiological relevance of GPR40/FFAR1 and of other FFA receptors [6,30], the mechanisms underlying the beneficial effects of DHA on the CNS, especially at the molecular level, are not fully understood. The identification of specific molecular pathways involved in the neurotrophic and neuroprotective actions of DHA may generate novel therapeutical strategies for CNS diseases. Here, we intend to summarize the experimental evidence and the mechanisms underlying the actions of DHA in neural cell survival, neuron morphology, and synaptic function at cellular and molecular level. In addition, we aim to provide molecular insights on the potential relevance of DHA on several CNS diseases and tumorigenesis. The relevance of signal transduction pathways, as well as the potential receptor systems involved, are highlighted and discussed.

### 2.1. DHA Actions on Neural Cell Survival

DHA has been recognized as one of the main FFA related to brain development [73]. A robust body of experimental evidence has shown the beneficial effects of DHA on the viability of cell lines of neural origin. Nevertheless, the molecular mechanisms underlying these positive DHA actions appear to be diverse (Figure 2A). Assays performed on neural progenitor cells (NPCs) showed that DHA (25–50 µM) stimulates the CREB pathway in a concentration-dependent manner, and increased the viability of NPCs in different stages, which included undifferentiated NPCs, cells at the initiation of differentiation, or cells differentiated from neuronal cells [74]. Interestingly, OLA did not induce any activation of the CREB pathway on NPCs [74]. These findings matched well with reports on other cell varieties, which showed that DHA incubation (25–50 µM) prevented cell death of cultured rat cortical neurons [31], inhibited the apoptosis of differentiated neurons from murine stem cells [75], and enhanced the cell viability of NGF-differentiated PC12 cells treated with a lipotoxic concentration of palmitic acid (PA) [76]. Further mechanistic insights showed that the protective effects of DHA on differentiated PC12 cells were related to a reduction in apoptosis and necroptosis through the reversion of the PA-induced enhancement of the activity of caspase-3 and caspase-8 [76]. Other authors explored the protective effects of DHA on PC12 using a model of oxidative damage induced by hydrogen peroxide (H_2_O_2_) [77]. These authors found that pre-treatment with 60 µM DHA (24 h) significantly antagonized the loss of cell viability induced by H_2_O_2,_ possibly through the regulation of the NFE2L2 (NRF2) antioxidant pathway [77]. Finally, investigations performed on Neuro-2A cells showed that the DHA-induced antiapoptotic effects were related to the activation of the Akt kinase and a subsequent increase in phosphatidylserine (PS) on the plasma membrane [78]. Additionally, these authors showed that the DHA deprivation in mice reduced hippocampal PS levels, which inhibited Akt signaling and increased apoptosis susceptibility in neuronal cell cultures [78]. These observations suggest that DHA promotes cell survival jointly through PS accumulation, Akt activation and the subsequent suppression of caspase-3 activity, leading to reduced cell death [78].

The effects of DHA on cell viability have also been examined in non-neuronal cells of the CNS, including Schwann cells (SCs) and microglial cell lines. Experiments with culture rat SCs showed that DHA (1 to 200 µM) had no effect on cell survival. However, DHA reversed the reduction in SC viability induced by PA by a mechanism involving the restoration of the Akt phosphorylation and mTOR pathway [79]. Further studies using immortalized mouse Schwann cells IMS32 also found protective actions of DHA (2.5–25 µM) on a model of cell toxicity induced by tert-butyl hydroperoxide (tBHP) [80]. Interestingly, DHA promoted the expression of anti-oxidative proteins (such as catalase and Ho-1), likely through an interaction with the nuclear factor NFR2 (Figure 2A) [80]. On the other hand, research focused on microglia have shown that high concentrations of DHA exerted negative effects on cell viability. Experiments on BV-2-immortalized mouse microglial cells showed that DHA displayed a negligible effect on cell survival at concentrations below 30 µM. However, higher concentrations (i.e., 100–200 µM) showed induction of cell death with noticeable morphological changes, including cytoplasmic swelling and size increase [81,82]. Similar experiments on BV-2 cells from other research groups also showed that DHA (>150 µM) generated detrimental effects on the cell viability [83]. Despite the evidence discussed above, research focused on DHA effects on astrocyte survival is lacking. Nevertheless, experiments performed with cultured astrocytes showed that DHA promotes its morphological differentiation [84]. Furthermore, other authors showed that DHA-treated astrocytes contributed to improve neuronal survival on astrocyte-neuron co-cultures [85].

### 2.2. DHA Effects on Neural Morphology

Mammalian neurons possess a large variety of morphological features [86,87,88,89]. Neuronal morphology has been historically established as one of the main forms of classification of all cell types of the CNS [90]. In neuronal cells, some essential morphological features are the soma, cell body size, dendrite number, axon length, and ramifications [91,92]. These parameters, among others, are frequently associated with functional and genetic data, allowing the morpho-functional classification of specific neuron subtypes [87,91,92].

Several studies have shown that DHA displayed positive effects in terms of morphological development of neural cells. For example, DHA treatment (10–30 µM) significantly accelerated neurite outgrowth of NGF-stimulated PC12 cultures [93]. These observations are in accordance with assays on cultured rat hippocampal neurons, which showed that culture medium supplemented with 1.5 µM DHA significantly enhanced total neurite length per neuron, individual neurite length, and the number of branches per neuron [94]. Similar results were also obtained on cultured mouse hippocampal neurons, where a sustained incubation with 1 µM DHA enhanced the number of branches and total neurite length per neuron [95]. An additional report on dissociated hippocampal neurons showed that a 4 µM DHA treatment increased dendritic arborization complexity [96]. Furthermore, 1 µM DHA increased neurite length, axon outgrowth, and axon elongation rate (approximately 1.5-fold) on rat cortical neurons [97]. On the other hand, two different research groups found that FFAs other than DHA, such as oleic acid (OA), arachidonic acid (AA), and docosapentaenoic acid (DPA), were unable to influence the neuronal morphology [94,95]. It is also worth mentioning that the DHA metabolite DEA has been characterized as an endogenous factor that modulates neural morphology [64,65]. DEA is generated from DHA in the brain and is present in mouse hippocampal tissue. In assays performed on cultured hippocampal neurons, 0.1 µM DHA was unable to exert changes in the neurite length, whereas DEA stimulated neurite growth from 0.01 µM. Interestingly, the positive effects of DEA on neural growth were fully absent in neurons from mice lacking GPR110, suggesting a critical role of this GPCR in the actions of DEA in the brain [65].

This specific association between DHA and beneficial effects on neuronal morphology has also been found using in vivo models that simulated sustained dietary FA deficiencies. Cultured hippocampal neurons from rats submitted to ω-3 FA-restricted diets showed shorter neurite lengths in comparison with cultures from animals with normal ω-3 FA ingestion [94]. Notably, after supplementing the ω-3 FA-deficient cultures with DHA, neurite length was restored to control values [94]. Qualitatively similar results were obtained using cultured mouse hippocampal neurons treated with DHA, OA, AA, and DPA. These findings showed that only DHA was capable of reestablishing the morphological parameters of neurons in ω-3 FA-deficient mice [95].

The mechanisms underlying the effects of DHA on neuronal morphology are complex, and different signaling cascades and effector proteins involved have so far been identified (Figure 2B). A DHA target that participates in neuronal morphological development is the retinoid X receptor (RXR), a ligand-activated nuclear hormone receptor that acts as a transcription factor [98]. Retinoic acid regulates the activity of genes involved in cytoskeleton remodeling, being a relevant factor in neuronal development [99]. DHA has been characterized as a transcription factor and an endogenous RXR ligand [100]. Despite this information, the available evidence shows that the DHA concentrations needed to drive morphological changes on hippocampal neurons were almost 10-fold smaller than the DHA concentrations necessary to activate RXR, suggesting that the trophic effects of DHA are likely not mediated by RXR [101]. Other research lines have proposed that DHA effects may involve the release of brain-derived neurotrophic factor (BDNF), a widely recognized neurotrophin controlling neuronal morphology and synapse formation [102,103]. The potential relationship between DHA, BDNF, and its receptor, TrkB, was initially reported in pregnant rats fed with FFA-deficient diets. This treatment showed a significant reduction in brain BDNF levels, as well as lower levels of phosphorylated TrkB and CREB [104]. Furthermore, these authors demonstrated a positive correlation between the phosphorylated TrkB and DHA levels in many brain regions, including frontal cortex and hippocampus [104]. On the other hand, DHA supplementation in postnatal stages promoted an increase in pro-BDNF and mature BDNF content on rat hippocampus [105]. Aside from RXR and BDNF, a study performed on rat neural stem cells (NSCs) suggested that the GPR40/FFAR1 pathway may contribute to DHA actions on neural morphology [106]. This study showed that GPR40/FFAR1 overexpression enhanced neurite length and increased the number of individual branches on DHA-incubated NSCs. Although DHA also exerted significant trophic actions on non-transfected control NSCs, the enhancement and activation of GPR40/FFAR1 signaling induced a greater improvement in the morphological parameters and stimulated intracellular Ca^2+^ mobilization [106].

### 2.3. DHA as Modulator of Synaptic Function

FFAs and DHA play critical roles in brain development, particularly on synaptogenesis and synapse maturation [12,107]. During synaptic development, pre- and postsynaptic factors should be organized, and an appropriate collection of signaling pathways must be finely orchestrated. Synaptic compartments remodeling is a critical process that needs to adjust the FFAs and DHA content in synaptic membranes, especially in dendrites and axons [108,109].

Several studies have explored the relevance of DHA in synapse formation and maturation (Figure 2C). At molecular level, DHA supplementation increased the expression of key pre- and postsynaptic proteins (i.e., PSD-95, synapsin-1, among others), and membrane phospholipids (i.e., phosphatidylserine and phosphatidylinositol) in the hippocampus of adult gerbils [110]. Similarly, DHA promoted the expression of the presynaptic protein synapsin-1 on cultured hippocampal neurons, increasing the number of synapsin-1 puncta in neurites [95]. In addition, these authors also showed that the levels of key postsynaptic proteins related to glutamatergic synapses (i.e., GluR1, GluR2, NR1, NR2A, and NR2B) were significantly increased by DHA [95]. In the same line, imaging analyses of dissociated hippocampal neurons showed that DHA raised the density and puncta size of Bassoon and Homer proteins, which are markers of pre- and postsynaptic sites, respectively [96]. These modifications also promoted a higher alignment of pre- and postsynaptic specializations, suggesting that DHA promotes the formation of more functional synapses [96]. In line with these observations, synaptic proteomic analyses of the brains of rats with DHA-restricted diets showed a significant reduction in the expression of 18 pre- and postsynaptic proteins, including syntaxin-1, Munc18, synapsin-1, Bassoon, dynamin-1, SV2, PSD-95, NR2B, among others [111]. Furthermore, using protein network analysis, these authors also identified the CREB pathway and caspase-3 signaling as the main networks involved in the proteomic changes observed in DHA-deficient animals [111]. Lastly, it should be considered that DEA can also exert significant effects in terms of the positive modulation of synapse maturation. For example, a low concentration of DEA (0.01 μM) significantly enhanced the number of synapsin and PSD-95 puncta of hippocampal and cortical neurons [64,65]. These synaptogenic effects were linked to GPR110 receptor activation by DEA and the subsequent enhancement of cAMP levels in neurons.

DHA-induced molecular changes on synaptic protein expression suggest a consequent enhancement of synaptic function. A direct correlation between molecular changes in synapsis with functional measurements was found in DHA-supplemented cultures of hippocampal neurons [95]. Using electrophysiological recordings, these authors showed that DHA-dependent increases in neurite length, branching, and number of synapsin-1 puncta were translated into an enhancement in the frequency and amplitude of total spontaneous synaptic currents. Interestingly, DHA treatment also increased the average amplitude of both glutamatergic and GABAergic synaptic currents, while only the frequency of glutamatergic synaptic currents was significantly enhanced [95]. This preferential modulation of glutamatergic synapses by DHA suggest that processes of synaptic plasticity could be sensitive to fluctuations in DHA levels in the brain. This notion was explored using electrophysiological recordings and hippocampus immunocytochemical analysis of DHA-deprived mice [95]. As expected from the results on cultured neurons, the long-term potentiation (LTP) elicited by high-frequency stimulation was impaired in slices from DHA-deficient mice. This functional disruption of LTP showed correlation with a significant reduction in critical postsynaptic proteins involved in glutamatergic synapses, such as NR1, NR2A/N, and GluR1 [95]. In line with these findings, other studies found that dietary supplementation with DHA induced the hippocampal LTP of young mice [112], and enhanced the maturation of cortical networks in vitro [96]. Nevertheless, the manipulation of the DHA levels in the brain is not always translated into functional outcomes or changes in excitatory synapses. For example, acute DHA treatment in rat hippocampal slices was not able to modify either LTP or LTD [113]. On the other hand, acutely applied DHA facilitated LTP, but not LTD, of cortico-striatal networks [114]. In addition, DHA supplementation to rats under chronic stress reversed the alterations on hippocampal GABAergic synaptic transmission, while glutamatergic synaptic function remained unchanged [115].

Besides the evidence discussed above, direct actions of DHA and other FFAs on ion channels add another layer of complexity to the regulation of synaptic function [66]. Electrophysiological assays have demonstrated that many key ion channels involved in synaptic function, such as GABAA receptors, nicotinic acetylcholine receptors, and two-pore domain K+ channels, are modulated by DHA and diverse FFAs [66,116,117]. Current evidence suggests that FFAs may modulate ion channels either directly, by interacting with protein channels, or indirectly, by altering the lipid bilayer properties [118]. However, whether these modulatory mechanisms have a neurophysiological or pathological relevance to the synaptic regulation exerted by FFAs is still largely unknown.

### 2.4. DHA Effects on CNS Disorders

Since DHA and other FFAs are necessary for the correct growth and function of the CNS, a low intake or restricted access will increase the risk of a variety of CNS disorders, including attention deficit hyperactivity disorder (ADHD), autism, bipolar disorder, depression, and suicidal ideation [119]. For example, in major depressive disorder it has been widely reported that ω-3 PUFAs produce an anti-depressive effect [120]. Although this effect is exerted by EPA rather than DHA, these two FAs must be administered in an EPA/DHA ratio higher than two in order to obtain optimal anti-depressive results [120]. Recent studies have shown that autism spectrum disorder (ASD) and attention deficit hyperactivity disorder (ADHD), two increasingly prevalent neurodevelopmental disorders, have been in a rise. This increase might be associated with a higher dietary intake of ω-6 and lower of ω-3 PUFAs [121]. In ADHD, lower levels of ω-3 PUFAs were found in membranes, and this was correlated with the behavioral and learning problems associated with patients who suffer from this condition [122]. For ASD patients, DHA and ω-3 PUFA deficiencies have also been reported [122,123], while in patents with borderline personality disorder, it was observed that a 1 g/day treatment with EPA reduced the severity of aggressive and depressive symptoms [124].

#### 2.4.1. DHA Effects on Neurodegenerative Diseases

As described in previous sections, DHA is involved in neurodevelopmental processes, including synaptogenesis, neuronal differentiation and axonal growth and regeneration [125], suggesting a potential link between DHA deficiency and neurodegenerative diseases [75,126]. There is evidence that, in both animal and human models, an inadequate intake of maternal ω-3 PUFA may lead to aberrant CNS development and function [127]. On the other hand, ω-3 PUFA supplementation during the prenatal and perinatal periods is a potential protective factor of neurodevelopmental disorders [128]. Various studies showed that DHA could improve learning and memory, and also reduce neuronal loss [129]. Nine separate observational studies suggest a possible link between increased fish consumption and a reduced risk of Alzheimer’s disease (AD) [130,131]. Low ω-3 PUFA levels may be associated with AD, in particular where there is low dietary fish or ω-3 PUFA intake and low DHA levels [132]. In this study, the authors observed that serum cholesteryl ester-eicosapentaenoic acid and DHA levels were significantly lower in all of the Mini Mental State Examination (MMSE) score quartiles of patients with AD compared with control values. Additionally, serum cholesteryl ester-DHA levels were progressively reduced with increasing severity of clinical dementia [132]. ω-3 PUFAs can not only decrease the generation of beta-amyloid (Aβ) peptide, but can stimulate its elimination [133], and DHA seems to ameliorate the memory-related learning deficits of AD [134]. Many studies have been conducted regarding the use of ω-3 PUFA supplementation to reduce AD development [135]. A meta-analysis study concluded that increases in daily DHA intake by 0.1 g/day reduced the risk of AD by 37% [136]. DHA reduction has also been suggested to have a role in cognitive decline and major psychiatric disorders [137], and there is also evidence that ω-3 PUFA may be relevant in the pathophysiology of depression [138]. Therefore, it is plausible that treatment with DHA and related ω-3 PUFA may represent a valuable tool in the management of neurodegenerative diseases [139].

Parkinson’s disease (PD) is a another highly prevalent neurodegenerative disease. PD primarily affects dopaminergic neurons located in the substantia nigra [140]. The use of substances that can modulate the inflammatory response to reduce PD progression is an area that remains to be further explored. In this subject, the neuroprotective effects of two DHA-derived resolvins, RvD1 and RVD2, have been investigated in experimental models of PD [141]. In a study that used a cellular model of PD, RvD1 inhibited the 1-methil-4-phenylpyridinium ion (Mpp^+^)-induced expression of proinflammatory mediators [142]. On the other hand, DHA supplementation induced an improvement in locomotor activity parameters on in vivo models of Parkinson’s disease (PD) induced by the neurotoxin 1-Methly-4-phenyl-1,2,3,6-tetrahydropyridine (MPTP) [143]. A clinical study showed that patients with PD and concomitant depression supplemented with ω-3 PUFA from fish oil (four capsules/day for 3 months) significantly reduced their scores in the Montgomery–Asberg Depression Rating Scale (MADRS), and improved their ratings in other mood scales as compared to the control group treated with vitamin E [144].

#### 2.4.2. ω-3 PUFAs and the Control of Obesity

A wide variety of studies have examined the potential beneficial actions of ω-3 PUFA supplementation on animal and humans with obesity and metabolic disorders (reviewed in [145,146]). Most of these studies have administered variable amounts and proportions of EPA/DHA and have focused their main analyses on classical metabolic parameters, such as plasma lipidic levels, glucose tolerance, body weight, and fat accumulation, among others. While the evidence from assays with rodents have shown consistent beneficial effects of ω-3 PUFA within diverse obesity-related parameters [145,146,147], meta-analyses revealed that ω-3 PUFA supplementation in humans either promoted a mild decrease in body weight [148] or did not reduce body weight significantly [149,150].

Despite these controversial results, studies performed on animals have suggested that some beneficial effects of ω-3 PUFA on obesity are related to hypothalamic modulation. Since it is well established that high-fat diets promote time-dependent hypothalamic inflammation and even neuronal death [151,152,153,154], ω-3 PUFAs may contribute to normalizing the hypothalamic function in obesity. In this context, a pioneering study found that a five-day intracerebroventricular treatment with ω-3 PUFA rapidly decreased the body weight and reduced the food intake of rats with diet-induced obesity [155]. Interestingly, these beneficial changes correlated with a significant reduction on the inflammatory mediators on the hypothalamus. In addition, ω-3 PUFA promoted regulatory changes in diverse hypothalamic signaling pathways and neurotransmitter systems. These observations, together with results from other groups, have strengthened the idea that the hypothalamus is a major site of action to explain the positive ω-3 PUFA effects against obesity [156,157,158,159,160].

The signaling pathways and molecular mechanisms modulated by ω-3 PUFA at the hypothalamic level are complex and are still not well defined. Nevertheless, the current evidence has shown that the administration of ω-3 PUFAs contributes to decreasing the hypothalamic expression of pro-inflammatory mediators (e.g., IL-10, TNF-α, IL-6). On the other hand, ω-3 PUFAs induce changes in the activity and expression of diverse signaling proteins in the hypothalamus, including AMP kinase, Akt, JAK/STAT, SOCS3, BDNF, and the GPR120 receptor. Finally, the expression of key hypothalamic neurotransmitters, such as POMC, CART, NPY, and MCH, was also modulated by ω-3 PUFAs. Altogether, these observations suggest that EPA and/or DHA supplementation may promote complex molecular changes in the hypothalamus impaired by obesity, contributing to normalizing the hypothalamic function and likely to reestablishing energy homeostasis in mammals.

#### 2.4.3. DHA Roles in CNS Tumors

The antitumoral effects of PUFAs have been widely reported [161,162,163]. Research articles using both in vitro and in vivo approaches have shown that DHA induces apoptosis and/or decreases proliferation in different types of tumoral cells [163,164,165,166,167,168,169,170]. The mechanisms underlying these effects include DHA incorporation into plasma membrane, lipidic peroxidation, eicosanoid metabolization, and effects on nuclear receptors [171,172,173,174,175].

CNS tumors represent around 2% of all malignant neoplasia [176], and according to the Central Brain Tumor Registry of the United States (CBTRUS), the most frequent brain tumors are meningioma (37.1%), pituitary tumors (16.4%), glioblastoma (14.7%), and malignant peripheral nerve sheath tumors (8.5%) [177,178]. Brain metastases are the most common intracranial tumors, exhibiting high morbidity and mortality rates [179]. Depending on primary tumor location, CNS tumors have a 2–27-month survival rate [127]. Because most of the anticancer agents currently used cannot cross the blood brain barrier effectively, or are easily inactivated in the brain tumor microenvironment [173], novel therapeutic approaches targeting brain tumors are greatly needed [180].

Diverse PUFAs and DHA inhibit metastasis in preclinical in vivo models, as well as in the invasion and migration of tumoral cell models in vitro [14,161,163]. Another reason for considering DHA as a promising alternative for CNS tumor treatment can be attributed to its strong ability to cross the blood brain barrier [181]. Moreover, combined therapies of DHA and antitumoral agents increase treatment efficacy because DHA increases reactive oxygen species (ROS) levels in tumoral cells, reducing their antioxidant defenses and increasing the absorption of antitumoral drugs [182]. Similarly, combined DHA/etoposide therapy shows efficacy in medulloblastoma affecting several signaling pathways. However, this combined therapy had no effect on glioblastoma [182]. ARA and DHA combination regulates proliferation, differentiation, and cellular migration through protein kinase C (PKC) in multiform glioblastoma-derived cells [183]. Another study evaluated the combined effect of DHA and lomustine, a lipid-soluble chelating agent that crosses the blood brain barrier, in the following CNS cancer cell lines: U87-MG (glioblastoma-astrocytoma), DB029 (grade 3 glioblastoma), and MHBT161 (glioblastoma primary culture). This combination inhibited cell growth only in DB029 and MHBT161 glioblastoma-derived cell lines, and slightly inhibited proliferation in primary human-derived cerebral cortex endothelial cells. These results suggest that this combination has a certain degree of selectivity towards glioblastoma [184]. Regarding radiotherapy on rat astrocytoma cells, 36B10 cells supplemented with GLA, EPA, or DHA showed changes in the fatty acid composition of their phospholipids and neutral lipids, which increased the sensitivity of this cell line to the cytotoxic effects of radiotherapy [185].

These data suggest that DHA may act as a chemosensitizer, enhancing the antiproliferative effects of traditional anticancer drugs on in vitro and in vivo models, and in ongoing clinical trials [173,186,187]. These properties highlight DHA as an interesting therapeutic agent against brain cancer, especially considering its role as an essential and harmless nutrient for CNS health [127]. Another advantage of FFA and DHA use in cancer therapeutics is their anti-inflammatory effects, which are, at least in part, due to their ability to act as agonists of the GPR40/FFA1 and GPR120/FFA4 receptors [6,30,67]. Inflammation has been recognized as a major hallmark in cancer growth and development [188,189], with chronic inflammation causing several degrees of damage to nucleic acids, proteins, and lipids through reactive oxygen and nitrogen species (ROS/RNS) generation. In a physiological context, affected tissues can activate stem cells for healing and regeneration. However, stem cells could also be damaged by ROS/RNS-induced inflammation, generating cancer stem cells (CSC) [190,191], and thereby promoting cancer development and inducing tumorigenesis [192]. This situation, combined with chemotherapy-induced damage, psychological stress, and high depression rates, portrays cancer as a pro-inflammatory scenario [189,193]. In this context, reports inform that ω-3 PUFAs can be metabolized into bioactive compounds beneficial for acute inflammation resolution [63,194]. Moreover, EPA and DHA can interact and activate GPR40/FFA1 and GPR120/FFA4 receptors inducing anti-inflammatory effects. Evidence also shows that GPR40/FFA1 and GPR120/FFA4 receptors have roles in tumorigenesis, migration, and metastasis [6,195,196,197,198].

Another aspect strongly related to cancer is pain. The generation and maintenance of cancer pain is related to multiple factors, including tumor invasiveness, cancer growth, neighboring tissue destruction and toxicity of chemotherapeutic agents [199]. These situations can induce acute and chronic pain, which are associated with sleeping difficulties, discomfort, and depression [200]. Together with its potential as an adjuvant against tumor growth, the activation of the GPR40/FFA1 receptor by DHA could also help to attenuate cancer pain. Cell signaling mediated by the GPR40/FFA1 receptor stimulates the descending pain-control system by inducing β-endorphin release [71,72], while its direct activation with DHA exerted anti-nociceptive effects in the formalin model of inflammatory pain [36]. In addition, DHA is directly involved in the biosynthesis of resolvins, protectins, and maresins, which are signaling molecules promoting anti-nociceptive effects at both peripheral and central sites [63,194].

## 3. Concluding Remarks

Current cumulative data undoubtedly validate the critical role of DHA and other FFAs in many aspects of neuronal physiology, including survival, morphological development, and synaptic function. The indications of the present data are strengthened by results from models of different levels of complexity, ranging from cell lines and neuronal primary cultures to brain preparations from mice with different access to dietary FFAs. Furthermore, results provided by researchers in the cancer field suggest that DHA and FFAs may also be effective as adjuvants against brain tumors.

Despite the growing number of reports, the molecular pathways underlying the beneficial effects of DHA and of other FFAs on the cells of the CNS are still a matter of debate. Although the direct interactions of DHA with membrane lipids and ion channels cannot be disregarded as potential mechanisms of action, robust experimental proof has shown that DHA may preferentially exert its actions through specific signaling cascades, likely involving the activation of receptors localized on the plasma membrane and in the intracellular environment. The existing results suggest that the BDNF/TrkB pathway, Akt signaling and the GPR40/FFA1 cascade are pivotal for at least some of the beneficial effects of DHA at the cellular level. Additionally, there is growing evidence regarding the effects of DEA, a metabolite of DHA that can signal through GPR110. Whether these pathways differentially modulate neural cell subtypes, or work synergistically with unidentified mediators, remains an open question for future research.

A deeper characterization of the signaling pathways involved in the DHA actions in neuronal physiology is required to better understand the critical roles of FFAs and DHA during brain development. Additionally, the discovery of such pathways may boost progress towards novel therapeutics against many CNS diseases, such as neurodegeneration and brain cancer. Of significant concern is the problem of determining the relevance of GPR40/FFA1 and other FFA receptors for the actions of DHA in physiology and disease. It is likewise still unclear whether the BDNF signaling pathway is necessary or sufficient to facilitate the effects of DHA on neuronal morphology and synaptic function. The discovery of improved synthetic agonists, antagonists, and allosteric modulators for both FFA receptors [67] and neurotrophin receptors [201] may soon boost new stimulating knowledge in the field. The combination of these pharmacological tools together with genetically modified mice may help to clarify the role of these signaling pathways in the beneficial but rather complex effects of DHA and other FFAs in the CNS.

## Figures and Tables

**Figure 1 ijms-23-05390-f001:**
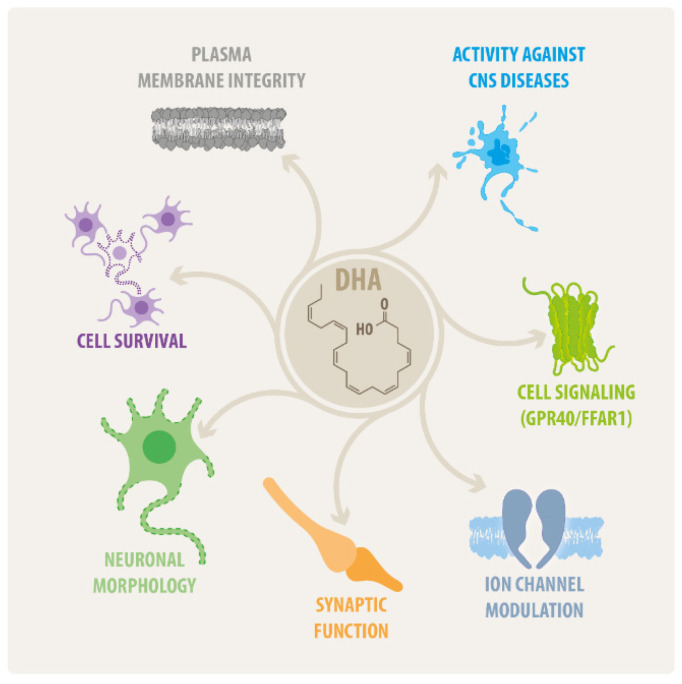
**DHA roles in the CNS.** Schematic representation of the main roles of DHA on the CNS, including structural, physiological, and therapeutic effects. Detailed information about these roles is properly developed on each section of this review.

**Figure 2 ijms-23-05390-f002:**
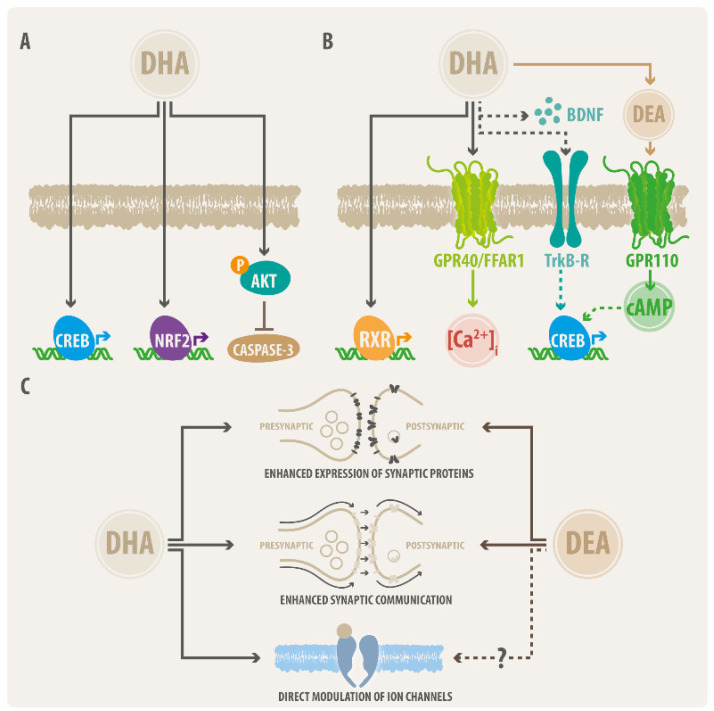
**DHA signaling pathways in neuron survival, morphology, and synaptic function.** (**A**) DHA can promote neuron survival by different mechanisms, including the induction of cAMP response element-binding (CREB) and nuclear factor erythroid 2-related factor 2 (NRF2) pathways, or apoptosis inhibition by reducing the activities of caspase-3 and caspase-8. (**B**) Regarding neural morphology, DHA can trigger the activation of retinoid X receptor (RXR) and the G-protein coupled receptor GPR40/FFAR1, which can increase intracellular calcium by activation of Gq/PLC/IP3 or by activation of the tyrosine kinase Trk-B receptor (Trk-B-R) by its activator, brain-derived neurotrophic factor (BDNF), with subsequent activation of the CREB pathway. This latter axis can also be activated by the DHA metabolite DEA (synaptomide), which induces a GPR110-mediated increase in cAMP levels. (**C**) The effects of DHA and DEA on synaptic function include the expression of synaptic proteins, such as postsynaptic receptors or scaffolding proteins (left scheme), which lead to an enhancement of synaptic communication (middle scheme). Additionally, DHA exert a direct effect on ion channels, modulating their gating properties (right scheme). Whether DEA is also able to modulate the activity of ion channels is unknown.

## Data Availability

Not applicable.

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
