# Peer review of "Roles of the Unsaturated Fatty Acid Docosahexaenoic Acid in the Central Nervous System: Molecular and Cellular Insights"

_ijms, 2022, doi:10.3390/ijms23105390_

Round 1
Reviewer 1 Report
Roles of the unsaturated fatty acid docosahexaenoic acid in the central nervous system
Ana B. Petermann, Mauricio Reyna-Jeldes, Lorena Ortega, Claudio Coddou and Gonzalo E. Yévenes
The manuscript summarizes and discuss the molecular mechanisms underlying the actions of DHA in physiological and pathological conditions in neural cells with a special focus on processes of survival, morphological development, and synaptic maturation.
This manuscript addresses a very important issue in the neurological and metabolic field. This is an excellent and well written manuscript, but some points should be further explored and justified.
- Hypothalamus is an important FA-sensing brain region involved in energy homeostasis. A section discussing the beneficial central effects and mechanism of DHA for protection of obesity and metabolic disorders should also be included.
- DHA as modulator of synaptic maturation for neurodevelopmental disorders? A section about the mechanisms of DHA to increase the risk to a variety of CNS disorders, such as ADHD, autism, and bipolar disorder should be included.
- The section about the DHA effects on neurodegenerative disease should be furthers develop and detailed. To poor.
- Page 2 line 49 – Use lowercase ω
Author Response
We appreciate the helpful comments of reviewer 1. Pleasse see attachment.

Reviewer 2 Report
This is an interesting review of DHA in the CNS. It is largely based on research findings from cell culture, but does also consider animal and human studies. There are several important functions of DHA which have not been considered and in addition the reference to free fatty acids (FFA) throughout is misleading in many respects (in cell membranes, dietary fatty acids etc).
This reviewer has a number of comments for the authors to consider.
- Lines 37, 38: Reference to “different types of FFA”. The literature refers to different types of fatty acids, and very rarely to FFA. Fatty acids in foods and in the body are mostly esterified and exist as FFA only after liberation from these lipids by lipases. It is acknowledged that in vitro studies use FFA as these are readily taken up into cells, but dietary studies largely use lipids such as triacylglycerol or phospholipids which contain esterified fatty acids.
- DHA is an omega 3 fatty acid – the authors use two different omega symbols – Ω and ω (eg. Line 334). Suggest use “ω “ as this is the more common of these two symbols.
- Throughout the ms the implication is that lipids exist as free fatty acids (FFA); it is false to give this impression (eg line 37). For example, DHA predominantly is found esterified to sn2-position in glycerophospholipids, not as a FFA. Some of the mechanisms of action of DHA will relate to its role in “fluidity’ of membranes (DHA as a component of phospholipids) in the facilitation the action of receptors; other actions will relate to DHA being a precursor of lipid mediators, presumably through PLA2 action to release the DHA (as a FFA) which can then act as a substrate for the various lipid mediators. It is not clear from the literature whether DHA as a FFA has a particular role.
- The authors should make this clear.
- The authors are referred to a recent review of ‘DHA in the brain – what is its role?’ (Sinclair AJ Asia Pac J Clin Nut 2019) which discusses the above and indeed many other matters missing from the present ms.
- The role of DHA in visual function is not referred to and visual function is intimately related to the CNS. This role of DHA is well established (see Sinclair 2019).
- The predominant role of EPA rather than DHA in major depression is not referred to (see Guu TW, et al International Society for Nutritional Psychiatry Research Practice Guidelines for Omega-3 Fatty Acids in the Treatment of Major Depressive Disorder. Psychother Psychosom. 2019;88(5):263-273. doi: 10.1159/000502652).
- The role of arachidonic acid as a major PUFA in brain phospholipids (PL) is not referred to; it is found in cell membrane PL together with DHA and in myelin it is present in higher amounts than DHA. Arachidonic acid has important roles in the CNS and it should at least be mentioned in the present ms (see Sinclair 2019; Bosetti F.J Neurochem. 2007).
- The authors refer to DHA and pain, but do not refer to several important intervention studies on chronic headache (line 83) (see papers such as Ramsden CE, et al. Sci Signal. 2017 and BMJ. 2021).
- Line 51: the authors failed to include DPA as a significant omega 3 fatty acids. Its proportions in the brain exceed EPA (see Sinclair 2019).
- Line 61: It is not clear what is mean by “less than 1% of total DHA”? Please clarify. Do the authors mean “reaching the brain”? Rapoport SI and Bazinet RB have reported that DHA synthesis in the liver in the rat from alpha-linolenic is able to supply neural DHA by a considerable factor!
- Line 81-83: The authors imply that the outcomes from interventions are consistently beneficial. This is misleading – “some studies have shown some benefits” in these conditions would be more consistent with the literature.
- Line 225 and 246: “ restricted FFA diets” and “…rats fed FFA-deficient diets”, respectively. These are meaningless statements. Diets contain lipids and mostly the fatty acids are esterified to glycerol in triacylglycerol. It is presumed the authors mean omega 3 deficient diets? Please specify to remove misleading statements.
- Line 281: “DHA restricted diets”? Do the authors mean omega 3 deficient diets?
- The authors do not refer to a DHA metabolite discovered by HY Kim and colleagues, and known as synaptomide. This compound has important properties in the CNS (see Sinclair 2019).
- The authors do not refer to the effects of omega 3 deficiency on neural glucose uptake, or to effects on Na/K ATP’ase activity, or to effects on olfaction and auditory function presumably indicating several additional important functions of DHA (see Sinclair 2019).
Author Response
We are grateful forthe helpful comments of reviewer 2. Please see attachment

Round 2
Reviewer 2 Report
The authors have carefully revised the manuscript.
One matter that has not been considered is that some of the effects of an omega 3 deficient diet might be due to exaggerated metabolism of arachidonic acid to PGE2 and related compounds. In particular, the reversal of learning defects associated with omega 3 deficiency was effected either by adding omega 3 fatty acids or by adding a COX inhibitor to drinking water of the animals. This should be discussed.
The text below outlines the evidence.
The authors should consider adding this to their comprehensive manuscript.
Learning defects in mice reared for 3 generations on omega 3 deficient diets could be reversed not only by the addition of omega 3 fatty acids (ALA) to the diet, but also by a COX-2 inhibitor [1].
This data might be explained by other data which reported that omega 3 deficiency significantly increased the activity, protein concentration and mRNA expression of the arachidonic acid regulatory phospholipase A2 ( PLA2) (calcium-dependent cPLA2 and secretory sPLA2), and COX-2 in rat frontal cortex [2] and the expression of cPLA2, COX-2 and PGE2 in the rat hypothalamus [3] thus presumably increasing the availability of free arachidonic acid for metabolism to PGE2.
[1] Hafandi A, Begg DP, Premaratna SD, Sinclair AJ, Jois M, Weisinger RS. Dietary repletion with omega3 fatty acid or with COX inhibition reverses cognitive effects in F3 omega3 fatty-acid-deficient mice. Comparative medicine. 2014;64(2):106-9.
[2] Rao JS, Ertley RN, DeMar JC, Jr., Rapoport SI, Bazinet RP, Lee HJ. Dietary n-3 PUFA deprivation alters expression of enzymes of the arachidonic and docosahexaenoic acid cascades in rat frontal cortex. Mol Psychiatry. 2007;12(2):151-7.
[3] Begg DP, Puskas LG, Kitajka K, Menesi D, Allen AM, Li D, et al. Hypothalamic gene expression in omega-3 PUFA-deficient male rats before, and following, development of hypertension. Hypertension research : official journal of the Japanese Society of Hypertension. 2012;35(4):381-7.
